# Hearing Loss and Cognitive Impairment: Epidemiology, Common Pathophysiological Findings, and Treatment Considerations

**DOI:** 10.3390/life11101102

**Published:** 2021-10-17

**Authors:** Antonella Bisogno, Alfonso Scarpa, Stefano Di Girolamo, Pietro De Luca, Claudia Cassandro, Pasquale Viola, Filippo Ricciardiello, Antonio Greco, Marco De Vincentiis, Massimo Ralli, Arianna Di Stadio

**Affiliations:** 1Department of Otolaryngology, University of Salerno, 84125 Salerno, Italy; a.bisogno91@gmail.com (A.B.); alfonsoscarpa@yahoo.it (A.S.); dr.dlp@hotmail.it (P.D.L.); 2Department of Otolaryngology, Tor Vergata University, 00133 Roma, Italy; Stefano.di.girolamo@uniroma2.it; 3Department of Otolaryngology, University of Turin, 10126 Turin, Italy; claudiacassandro@inwind.it; 4Unit of Audiology, Department of Experimental and Clinical Medicine, Magna Graecia University, 88100 Catanzaro, Italy; pasqualeviola@unicz.it; 5Department of Otolaryngology, Cardarelli Hospital, 80131 Napoli, Italy; Filippo.ricciardiello@aocardarelli.it; 6Department of Sense Organs, Sapienza University of Rome, 00161 Rome, Italy; antonio.greco@uniroma1.it (A.G.); massimo.ralli@uniroma1.it (M.R.); 7Department of Oral and Maxillofacial Sciences, Sapienza University of Rome, 00161 Rome, Italy; marco.devincentiis@uniroma.it; 8Otolaryngology Department, University of Perugia, 06129 Perugia, Italy

**Keywords:** hearing loss, dementia, hearing aid, cognitive impairment, Alzheimer’s disease, epidemiology, risk factors, pathophysiology

## Abstract

In recent years, there has been increasing research interest in the correlation between hearing impairment and cognitive decline, two conditions that have demonstrated a strong association. Hearing loss appears as a risk factor for cognitive impairment, especially among certain populations, notably nursing home residents. Furthermore, hearing loss has been identified as a modifiable age-related condition linked to dementia, and it has been estimated that midlife hearing loss, if eliminated, might decrease the risk of dementia in the general population. Several mechanisms have been suggested to explain the pathologic connections between hearing loss and dementia; however, clear evidence is missing, and the common pathophysiological basis is still unclear. In this review, we discussed current knowledge about the relationship between hearing loss and dementia, and future perspectives in terms of the effects of hearing rehabilitation for early prevention of cognitive decline.

## 1. Introduction

Hearing loss and cognitive impairment are conditions with a significant incidence in the ageing population, in which they often coexist with mutual influences. Available evidence has shown that the risk of dementia is higher in patients with hearing loss compared to healthy controls [1,2,3], and that in older adults with age-related hearing loss (ARHL), accelerated brain changes and reduced volume of auditory cortex are present [4,5].

This correlation between dementia and hearing loss emerged in the late 1980s, with a case-control study that found that cases of Alzheimer’s disease (AD) had twice the prevalence of hearing loss compared to controls, and a greater degree of hearing loss corresponded to a higher risk of developing severe cognitive decline [6].

Additional investigations conducted in the following years confirmed this strong association. Lin et al. monitored with pure tone audiometry for 12 years over 600 older subjects with no diagnosis of dementia. The authors found that the presence of dementia was significantly higher in patients with hearing loss compared to people who had no hearing impairment [7]. Gallacher et al. monitored a group of 1057 men for 17 years and found that the risk of developing dementia was nearly three times higher in patients with hearing loss [8]. Similar results were obtained in other consequent studies, that also demonstrated a greater decline in cerebral activity of people with hearing loss [9,10,11,12] (Table 1).

ARHL has been linked not just to cognitive impairment, but also to other conditions such as depression, falls and frailty [5,14]. Dementia and hearing loss are both independent risk factors for delirium, and their effects are likely to interact, especially in noisy environments such as hospitals [15]. Furthermore, hearing loss increases the likelihood of all forms of psychotic symptoms [16], and hallucinations in patients with dementia occur more commonly in people with sensory impairments, both visual and auditory [17].

The aim of this review is to highlight and discuss the close relationship between hearing loss and cognitive impairment and explore the known causal mechanisms; furthermore, this review examines the importance of preserving physiological hearing function and of promptly initiating the necessary hearing rehabilitation to prevent cognitive decline.

## 2. Materials and Methods

The systematic review was performed following PRISMA (Figure 1).

### 2.1. Search Strategy

Two researchers independently conducted literature research on PubMed and Scopus. The search terms were: “hearing loss”, “dementia”, “cognitive decline”, “Alzheimer disease”, “hearing impairment”, “comorbidities”, “brain change”, “brain atrophy”, “auditory screening” and “hearing rehabilitation” as well as their possible combination.

### 2.2. Inclusion and Exclusion Criteria

Only articles in the English language were considered. Any type of study (CRTs, Meta-analyses, and Systematic reviews) was included if discussed correlation/association between hearing loss (HL) and cognitive impairment; association/correlation between HL and AD; association/correlation among HL, cognitive impairment and systematic disease; description of common pathophysiological mechanism among HL and cognitive impairment, HL and AD, HL and cognitive impairment and AD; treatment of HL and effect on cognition/memory; treatment of HL and effect on cognitive impairment/AD outcomes. 

All review articles were also read in full. The related citation tool in PubMed was used to search further potential articles. References of all selected articles were also examined to identify additional relevant publications and ensure all applicable literature was included.

Articles in French, Spanish, Chinese or other languages were excluded. Case report, short communication, letter, opinion, and editorial were excluded.

### 2.3. Analysis of the Articles

After the removal of duplicates, 250 articles were identified. The title and abstract of each article were reviewed by the authors and the full-text article which contents corresponded to inclusion criteria were read in full by both researchers. Ninety-five articles containing the relevant info focused on answering the question about the impact of central/peripheral HL on CI and brain changes were used to prepare this systematic review.

## 3. Epidemiology

Hearing loss and cognitive impairment are widely diffused conditions in the ageing population. The burden of hearing loss is constantly growing for the increased exposure to loud sounds during lifetime and for the increase in life expectancy. Actually, the global number of people with hearing loss is estimated at 360 million by the World Health Organization (WHO) and this number is expected to double to 720 million by 2050 [18]. In the world, one-third of the population over 65 suffer from disabling hearing impairment (i.e., a hearing loss over 40 dB in the better hearing ear) and this number may reach over 900 million people by 2050 [18].

Dementia is the greatest global challenge for health and social care in the 21st century. It occurs mainly in people older than 65 years and its incidence is also steadily rising because of increasing longevity [18,19]: currently, there are around 50 million people living with this condition worldwide and this number is likely to grow by at least three times to nearly 152 million by 2050, with more than half of people aged 85 or older affected by cognitive impairment [19].

It is therefore evident that both hearing loss and cognitive impairment are age-related conditions that often coexist, especially among certain groups of the population, notably nursing home residents [20].

Hearing loss and cognitive impairment also have a huge economic impact. WHO estimates that untreated hearing loss poses an annual global cost of US$ 750 billion [18] and the current annual cost of dementia is about USD 1 trillion, predicted to double by 2030 [18]. The impressive impact on patients and their families, which cover nearly 85% of costs [20,21], is worrying to the point that prevention and treatment of dementia are a healthcare priority throughout the Western world [21].

## 4. Definition of Hearing Loss and Cognitive Deficit in Elderly Population

Sensorineural hearing loss (SNHL) in the elderly is generally defined as age-related SNHL. Different mechanisms cause the auditory deterioration and, depending on the portion of the auditory pathways affected, can appear in different forms [22,23].

SNHL can be classified as peripheral in case of damage limited to the cochlea or to the cochlear nerve [22]; differently, central SNHL is referred to the damage at the level of the superior auditory pathways (cochlear nuclei and auditory cortex [24]. In many cases, especially for ARHL, both peripheral and central involvement may be present.

In those cases, in which SNHL is caused by damage of the peripheral auditory pathways, three forms of presbycusis can be defined according to the structures involved.

*Sensory presbycusis* results from the degeneration of the organ of Corti, in particular from the deterioration of outer hair cells starting from the basal turn of the cochlea (high frequencies) to the apex (low frequencies) [25,26].

*Neural presbycusis* is referred traditionally to as the loss of spiral ganglia and to the damage of the cochlear nerve [27] with potential cochlear synaptopathy [28]. In this case, patients present a moderate downward slope in the pure tone threshold for high frequencies and a severe decrease in speech discrimination compared to the pure tone threshold [29]. Speech discriminations decrease by the increase in spiral ganglia cells loss of over 50% [30].

Finally *Atrial or metabolic presbycusis* results in hearing loss across the entire frequency range in the audiogram. It is caused by atrophy of the vascular stria, in which the loss of 30% or more of the tissue in the vascular stria causes a decrease in the hearing threshold [30].

Central SNHL is caused by the alteration of the superior auditory pathways (cochlear nuclei, auditory cortex). These structures can be affected by vascular damage [31,32] that alters the signal transmission. Brain atrophy, which could be a consequence of vascular damage or aging is another potential cause of central SNHL [24].

Brain atrophy is one of the links between cognitive decline, Mild Cognitive Impairment (MCI), Alzheimer’s Disease (AD) and hearing loss; in fact, all conditions cause atrophy of the brain [33].

Cognitive decline is defined as the reduced cognitive function consequent to age [34], when the cognitive decline is more severe than expected for the person’s age MCI is diagnosed [34]. Finally, AD is a progressive neurodegenerative disease marked by deficits in episodic memory, working memory (WM), and executive function, which has specific biomolecular markers [35]. MCI seems to be an early sign of AD [34].

In these conditions, memory (both working and short) progressively declines. In addition, psychological diseases can further impact memory function, and the elderly are more exposed to this problem [36].

In this review, we refer to general cognitive decline (impairment), without a clear focus on MCI and AD.

## 5. Common Pathophysiological Mechanisms

Today, hearing loss is considered as the most relevant modifiable risk factor for dementia, that, if eliminated, could reduce the risk of dementia by 9% [37]. The youngest mean age in which the presence of hearing loss was shown to increase dementia risk is 55 years [8]; several cohort studies [22] have demonstrated that even mild levels of hearing loss increase the long-term risk of cognitive decline in individuals who are cognitively intact at baseline.

The underlying causal mechanisms leading to the connection between these two conditions are not well understood, though several possible mechanisms have been suggested [38]. Generally, the outcome is a sort of bidirectional vicious circle, in which on the one hand hearing loss involves structural and functional changes to the brain, and on the other cognitive decline correlated with age facilitates the onset of hearing deficiency and entails a loss of perception and verbal comprehension (Figure 2).

Several theories have been proposed to explain the interactions and common pathophysiological mechanisms of cognitive decline and hearing impairment. They include increased cognitive load, changes in brain structure and function, common genetic and pathologic causes, and social disengagement.

### 5.1. Increased Cognitive Load

The cognitive load, which theory was developed in 1998 by psychologist John Sweller [39,40], consists of the cognitive effort or amount of information processing necessary to perform a specific task: if a learning task requires too much effort, learning will be affected because cognitive capacity in working memory is limited [38]. According to this hypothesis, hearing loss probably increases the cognitive effort required to process and understand speech, since reduced or distorted sensory input will require the brain to work harder [41]. An excessive cognitive load dedicated to auditory perceptual processing in everyday life could cause relevant brain structural changes and neurodegeneration to the detriment of other cognitive processes such as working memory, with the consequent creation of a vicious cycle in which the cognitive resources available for auditory perception may be reduced: hypothetically, this could lead to cognitive decline, but scientific evidence for this hypothesis to the present moment is limited [38]. In particular, the results of a systematic review performed by Ohlenforst et al. [42] indicated that studies on the theme lack consistency and have insufficient statistical power, although in hearing-impaired patients electroencephalographic (EEG) response to acoustic stimuli showed higher listening effort than healthy controls. Instead, in another review by Van Engen and McLaughlin [43], pupillary responses measured with pupillometry and eye tracking in subjects with hearing loss seem to indicate greater cognitive load as speech becomes less intelligible.

### 5.2. Changes in Brain Structure and Function

Nowadays, there is evidence that hearing impairment is associated with cerebral alterations. Specifically, magnetic resonance imaging studies demonstrate that hearing loss is correlated with reduced volume of the whole brain and of the primary auditory cerebral cortex in the temporal lobe [5,38] (Figure 3).

The arterial spin labelling magnetic resonance imaging, instead, analyses cerebral perfusion in vivo and allows non-invasive investigation of brain perfusion changes. This technique revealed that perfusion within the bilateral primary auditory cortex in patients with hearing impairment is reduced, especially in the right lateral Heschl’s gyrus, a critical area for auditory processing and which is also involved in many other cognitive skills [44].

Di Girolamo et al. investigated differences in sound stimulus processing and in connectivity of the primary auditory cortex in 131 patients affected by AD and in 36 normal subjects with brain PET/CT, finding reduced glucose consumption in brain areas 6, 7, 8, 39, whereas no differences were found in the primary auditory cortex [45].

Chronic hearing impairment also induces less activation of the central auditory pathways, a dysfunction of the auditory–limbic pathway and atrophy of the frontal lobe [46] and of the hippocampus [47].

In parallel, hearing impairment is associated with an increase in stimuli coming from other sensory organs, such as the eyes. This can lead to a “compensatory” increase in volume in other areas, following cross-modal cortical reorganization, which reflects the brain’s ability to compensate for alterations/dysfunctions of other senses through neuroplasticity mechanisms. Essentially, the brain adapts to a loss by compensating through neuroplasticity mechanisms; however, this phenomenon can have a seriously detrimental effect on cognition. In fact, in people with hearing loss, the compensatory adaptation system significantly reduces the brain’s ability to process sounds, which in turn affects a person’s ability to understand speech and, even with mild hearing loss, the hearing areas of the brain become weaker. Successively, the areas of the brain that are necessary for higher-level thinking compensate for the weaker areas, essentially taking over for hearing and leaving them unavailable to do their primary job [48]. This process would explain why people with hearing loss exhibit reduced cognitive performance, especially regarding executive rather than linguistic functions, during neuropsychological assessments.

### 5.3. Common Pathological Conditions 

Some researchers suggest that a common pathological condition might trigger both hearing loss and cognitive impairment: according to this hypothesis, both conditions are the results of a common neurodegenerative process in the aging brain [49,50,51,52,53], involving degeneration of the stria vascularis, loss of hair cells and primary afferent neurons, changes in neurotransmitter release [54].

Both ARHL and cognitive impairment are multifactorial and heterogeneous, with several common risk factors, mainly vascular, such as atherosclerosis, smoking and diabetes, that increase the risk of cardiovascular disease and stroke [31,37,55,56,57,58,59,60,61], as observed in a population-based survey conducted during 1997 to 1999 and 2002 to 2004 [62]. At the same time, the brain is susceptible to oxidative stress, which also plays an important microcirculatory role in auditory processing [54]. The role of the apolipoprotein E (APOE) gene has also been suggested, known to be strongly associated with neurodegeneration: some studies have also evidenced an association between the apolipoprotein E e4 allele, which predisposes to Alzheimer’s disease, and hearing loss, although the data in this regard are still fragmented [63,64,65] and the specific molecular link between hearing loss and dementia remains unknown [66]. In the same way, syndromic associations of dementia with dysfunction of the cochlea or ascending auditory pathways are uncommon and generally occur in the context of more complex neurological impairment, often in younger patients [67].

### 5.4. Social Disengagement

Social isolation might be another mechanism underlying the connections between hearing loss and cerebral alterations. In fact, social interactions are usually less satisfactory for individuals with hearing impairment because it is difficult for these patients to filter out a conversation from background noise. Communication difficulties associated with hearing loss can therefore encourage solitude, which is considered a risk factor for cognitive disorders, reduced cognitive stimulation, apathy and possibly depression [68], as shown by the correlation between greater hearing loss and increased odds of social isolation in a US representative sample of women aged 60 to 69 years [69], or by the residents of community settings and nursing homes with severe hearing loss, that show 1.4 times greater odds of demonstrating low social engagement and 1.3 times greater odds of spending little time participating in facility activities [70]. Depression associated with hearing loss and cognitive impairment has episodic nature and often manifests more in somatic symptoms and lethargy (so-called “depression without sadness”), therefore it may be very difficult to diagnose [46].

Social isolation also promotes negative biological mechanisms, such as increased transcription of pro-inflammatory genes and, therefore, an increase in the general inflammatory status, which is a major risk factor for damage even to cerebral functions [71,72].

## 6. Evaluation of Hearing Loss in Patients with Cognitive Impairment

According to the National Institute on Deafness and Other Communication Disorders (NIDCD), patients with hearing loss averagely wait 7 years before seeking treatment. During this time, hearing loss may affect cognitive function leading to an irreversible decline; furthermore, hearing loss is often not detected and/or not treated in people with a diagnosed cognitive impairment [68].

Conversely, performances on the cognitive tests that are used in the assessment of possible dementia are affected by various factors, including sensory impairment [73,74]. If a person is not able to hear the question that is being asked, he is obviously less likely to supply a correct answer. This becomes a problem when assessments are performed in environments with background noise, such as hospitals. Thus, it would be possible to misdiagnose a person with normal cognition as having cognitive impairment.

NICE guidelines on hearing loss recommend performing hearing evaluation for each patient with a clinical suspicion of cognitive impairment and should be repeated every two years in subjects with a diagnosis of dementia [75].

Moreover, recent evidence has shown that also central auditory system dysfunction occurs in the prodromal or early stages of AD; its screening is recommended as a low-cost and effective means to identify precursors to AD pathology, especially in at-risk populations [76,77,78].

Central hearing loss is characterized by difficulty in making sense of speech against noise [79], not explained by cochlear (peripheral) hearing impairment and therefore not susceptible to improvement with peripheral amplification (such as hearing aids) [56]. It is a condition whose prevalence increases with age; it may follow different brainstem and cortical lesions involving the auditory pathways and may coexist with peripheral hearing loss [80]. Patients with central hearing loss may perform normally on conventional hearing tests such as pure tone audiometry; however, they show an impaired speech perception [68]. This condition can be associated with hidden hearing loss, in which patients show a clinically normal audiogram but a worse perception of noise probably due to damage of the inner hair cells and spiral ganglion neurons [28,81,82,83,84,85].

Central hearing loss appears to be commoner in mild cognitive impairment and AD than in age-matched controls [80]: deficits of central auditory processing may therefore appear as potential biomarkers for neurodegeneration [86], though as of today there is little research that has linked hearing loss with the conventional biomarkers for AD [87].

## 7. Treatment of Hearing Loss in Patients with Cognitive Impairment

Growing evidence confirms that the treatment of hearing loss through hearing solutions is an effective solution to delay the onset of cognitive impairment and maintain a good cerebral function. Treating hearing loss in people with cognitive impairment may help improving communication, increasing quality of life, and reducing behavioral symptoms of dementia, with reduced stress for families and caregivers [7]. It is believed that these positive effects are achieved in various ways: if the increased cognitive load hypothesis is valid, hearing enhancement should reduce the effort required for auditory processing, redirecting it to cognitive tasks and then reducing the progression of brain atrophy; contrariwise, if the common pathological conditions hypothesis is valid, the development of cognitive impairment should progress independently from hearing intervention [23].To date, the treatment of hearing loss in patients with cognitive decline is based on three possible options: hearing aids, amplification devices and cochlear implants, whose impact remains unclear, while prevention is based on the use of local or systemic antioxidant therapies that have shown protective properties in contrasting the mechanisms of oxidative stress that lead to hearing loss [88,89,90,91,92,93,94,95,96].

### 7.1. Hearing Aids

The hearing aids are mostly used for treating hearing loss among people with cognitive impairment [87]: they seem to be more used by people in better health and of higher socioeconomic status [51] and there is mounting evidence of their efficacy to improve cognitive function in this population [97]. In a randomized trial conducted during a 4-month period, people using hearing aids showed a small but significant improvement in their cognitive performance as assessed by the Short Portable Mental Status [98]. Similar results underlining higher scores on cognitive tests after the use of hearing aids, also emerged from the study by Lin et al. [76], analyzing data coming from 605 hearing-impaired subjects between 60- and 69-years old. The cognitive benefit of hearing aids was later confirmed in a larger sample of people over 65 years of age, constituted by 1276 individuals, of which 137 with major hearing loss and 1139 with moderate problems as difficulty following the conversation when several persons talk at the same time or in a noisy background [11]. More studies [12,99,100] agreed in demonstrating a reduction in the rate of cognitive decline associated with hearing aid use; while the study by Dawes et al. [13] on 666 community-dwelling older adults with hearing impairment shows no evidence that hearing aids promote cognitive function, mental health, or social engagement.

### 7.2. Hearing Amplification Devices 

Other interventions, such as hearing amplification devices and focused communication strategies, have been applied in both in-patient [101] and outpatient settings [102] with some evidence of benefit on outcomes such as communication and depression. In their systematic review, Shukla et al. [79] found higher scores in cognitive tests and improved perception of communication of older hearing-impaired patients after the use of a voice amplifying device; beneficial results were also confirmed by Mamo et al. [80] and extended to depression, neuropsychiatric and behavioral symptoms.

### 7.3. Cochlear Implant

Different from conventional hearing aids, cochlear implants can restore hearing in patients with profound hearing loss [103]. Nevertheless, there is so far little research on the use of these devices to treat severe to profound hearing loss among people with cognitive impairment or dementia [104,105,106,107,108]. Studies available to date, such as that performed by Mosnier et al. [86] in 10 French tertiary referral centers, suggest an improvement in attention, episodic and working memory, and processing speed in cochlear implants users, even after as little as 6 months from implantation [97].

Despite promising results, studies on cochlear implants and cognition still have some important limitations. First, the long-term effects of cochlear implants on cognition are still unclear since most studies simply repeat neurocognitive tests at 6 and 12 months after implantation. Second, many of these studies did not have control groups, except that of Jayakody, comparing 16 implanted adults to 23 candidate adults waiting for surgery [105]. Third, inappropriate use of measures that are delivered through hearing may overestimate cognitive dysfunction in subjects with severe hearing loss. Last, the mechanisms that underlie changes in cognitive function following cochlear implantation are still mainly underexplored [97].

## 8. Conclusions

Available evidence confirms that hearing impairment is an independent and modifiable risk factor for cognitive decline. Our group already showed that hearing rehabilitation could positively impact the elderly brain function [24]; in fact, the amelioration of hearing positively improves both attention and memory functions [109]. Progressive limitation in cognitive skills, functional independence and social relations have been shown in patients with cognitive decline and hearing loss; hence, the need to strengthen research that investigates the connections between the two conditions is necessary to find adequate clinical answers.

In this perspective, it is essential to understand the mechanisms underlying the correlation between hearing and cognition, to prevent the onset of hearing disorders and, hence, of cognitive impairment. At the same time, it is equally essential to promptly recognize hearing loss; indeed, intervening to solve the problem means implementing a genuine therapy against cognitive decline with huge benefits for the individual person and society.

## Figures and Tables

**Figure 1 life-11-01102-f001:**
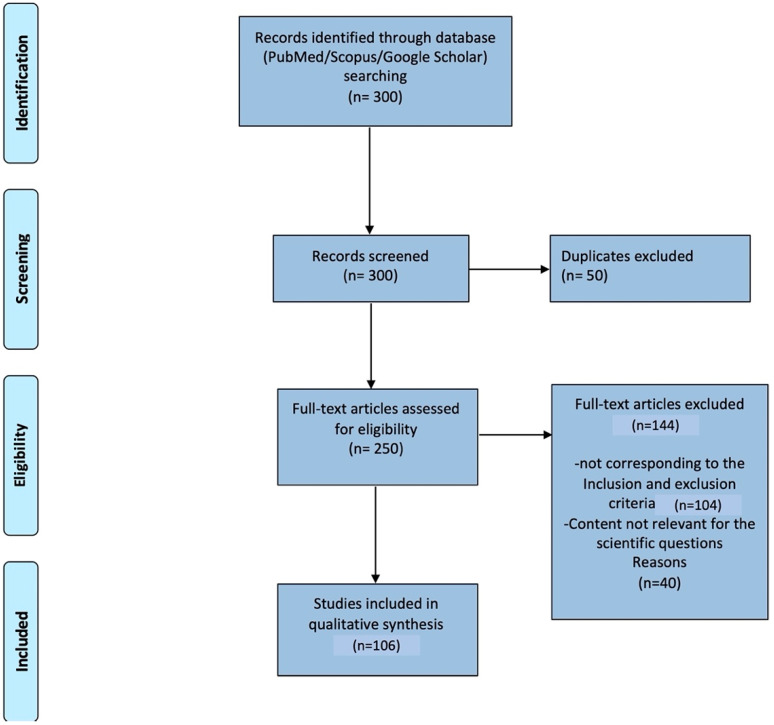
Prisma Flow shows how we selected and extracted the articles.

**Figure 2 life-11-01102-f002:**
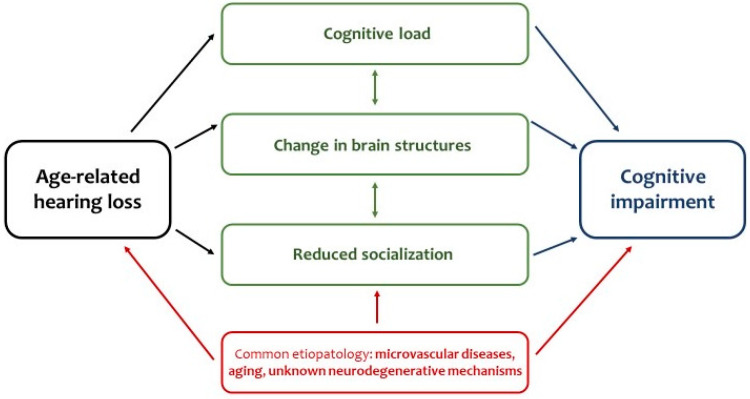
Causal connections between hearing loss and cognitive impairment.

**Figure 3 life-11-01102-f003:**
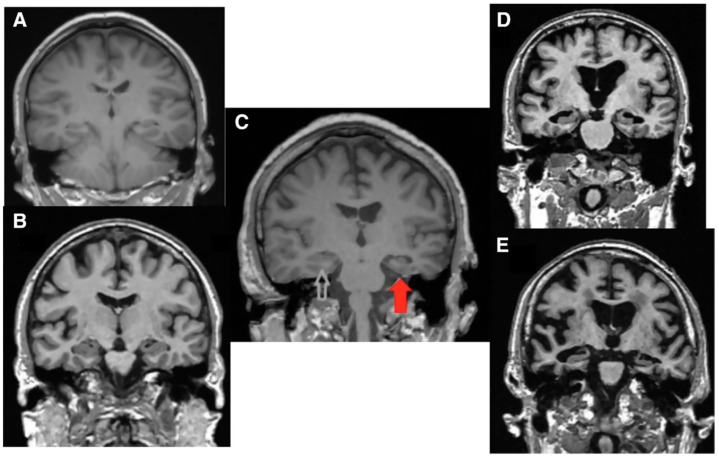
Magnetic resonance imaging, coronal view. (**A**) Normal brain in a young subject; (**B**) Normal brain in an elderly subject; (**C**) Temporal brain atrophy in a 68-year-old patient with asymmetric severe hearing loss (red arrow). The white arrow shows the normal temporal lobe; (**D**) Diffused brain atrophy in a 72-year-old patient affected by cognitive decline; (**E**) Diffused brain atrophy in a 65-year-old patient with Alzheimer’s Disease [5].

**Table 1 life-11-01102-t001:** Studies on the association between hearing loss and cognitive impairment.

Author and Year	Population Studied	Results
Uhlmann, 1989 [6]	100 cases with Alzheimer’s dementia and 100 nondemented controls	Hearing loss of 30 dB or greater significantly higher in cases than in controls. Greater hearing loss associated with a more severe dementia.
Lin, 2011 [7]	639 older adults without dementia	Mild, moderate or severe hearing loss associated with a risk of cognitive decline, respectively, two, three and five-fold higher than in normal-hearing population
Gallacher, 2012 [8]	1057 men	Risk of developing dementia 2.7-fold higher for every 10 dB of hearing loss
Amieva, 2015 [11]	3670 people aged 65 and older	Worse hearing associated with lower cognitive efficiency scores and with greater decline in cerebral activity
Dawes, 2015 [13]	164,770 adults aged 40–69	Frail hearing associated with higher levels of cognitive impairment
Meusy, 2016 [9]	600 men and women without dementia	Hearing loss in 77% of dementia population. ARHL associated with an over 3-fold increase in the probability of manifesting dementia
Fritze, 2016 [10]	154,783 people aged 65 and older, including 14,602 cases of dementia	Bilateral hearing loss associated with a 43% increase in the probability of developing dementia; unilateral hearing loss with a 20% increase

## Data Availability

Data are available under request to Alfonso Scarpa (alfonsoscarpa@yahoo.it).

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
