# Peer review of "Hearing Loss and Cognitive Impairment: Epidemiology, Common Pathophysiological Findings, and Treatment Considerations"

_life, 2021, doi:10.3390/life11101102_

Round 1

Reviewer 1 Report

This manuscript promises much, on a highly topical field of the contribution of hearing loss to cognitive impairment and decline. The authors have done a systematic search and found among the more appropriate papers. unfortunately not much is done in terms of analysis of these papers. There are generally no more than 1 line statements of what a paper found. The end result is a very superficial review of the field, and more driven by the authors' interests than any desire to advance the field. For that reason, I am not enthusiastic of the impact of the article on the field.

Author Response

Thank you for the careful review of our manuscript titled, “Hearing loss and cognitive impairment: epidemiology, common pathophysiological findings, and treatment considerations.”

We are grateful for the opportunity to submit an updated manuscript, and we have revised the paper in accordance with these valuable recommendations.  We thank the reviewers for their service and expertise, which has allowed us to materially improve the content and clarity of the submission.

We have addressed all critiques and discussed all changes in the responses below in bolded Italic. We used track change to show our in-text edits, and we have uploaded both a revised and a clean copy of the manuscript.

Reviewer 1

This manuscript promises much, on a highly topical field of the contribution of hearing loss to cognitive impairment and decline. The authors have done a systematic search and found among the more appropriate papers. unfortunately not much is done in terms of analysis of these papers. There are generally no more than 1 line statements of what a paper found. The end result is a very superficial review of the field, and more driven by the authors' interests than any desire to advance the field. For that reason, I am not enthusiastic of the impact of the article on the field.

Thank you for your comments. We did an exhaustive review of the literature and we included additional information. We also included new information and data, with related references, that could be interesting for readers to make the paper more appealing, and better discussed available evidence.

Reviewer 2 Report

This is a review on the correlations between hearing loss and cognitive impairment.

The manuscript is well written however I have the following recommendations

1 Better define the differences between peripheral and central age related HL

2 Better define cognitive impairment and differentiate between Dementia, mild cognitive impairment and subjective memory complain

3. Adjourn the references and take in consideration for example the following papers

1: Sardone R, Castellana F, Bortone I, Lampignano L, Zupo R, Lozupone M, Griseta C, Dibello V, Seripa D, Guerra V, Donghia R, Logroscino G, Solfrizzi V, Quaranta N, Ferrucci L, Giannelli G, Panza F. Association Between Central and Peripheral Age-Related Hearing Loss and Different Frailty Phenotypes in an Older Population in Southern Italy. JAMA Otolaryngol Head Neck Surg. 2021 Jun 1;147(6):561-571. doi: 10.1001/jamaoto.2020.5334. PMID: 33570584; PMCID: PMC7879383.

2: Sardone R, Battista P, Donghia R, Lozupone M, Tortelli R, Guerra V, Grasso A, Griseta C, Castellana F, Zupo R, Lampignano L, Sborgia G, Capozzo R, Bortone I, Stallone R, Fiorella ML, Passantino A, Giannelli G, Seripa D, Panza F,
Logroscino G, Quaranta N. Age-Related Central Auditory Processing Disorder, MCI, and Dementia in an Older Population of Southern Italy. Otolaryngol Head Neck Surg. 2020 Aug;163(2):348-355. doi: 10.1177/0194599820913635. Epub 2020 Apr 21. PMID: 32312167.

3: Lozupone M, Sardone R, Donghia R, D'Urso F, Piccininni C, Battista P, Di
Gioia I, Resta E, Castellana F, Lampignano L, Zupo R, Bortone I, Guerra V,
Griseta C, Seripa D, Solfrizzi V, Giannelli G, Quaranta N, Logroscino G, Bellomo A, Panza F. Late-onset depression is associated to age-related central auditory processing disorder in an older population in Southern Italy. Geroscience. 2021
Apr;43(2):1003-1014. doi: 10.1007/s11357-020-00290-1. Epub 2020 Oct 31. 

Author Response

Thank you for the careful review of our manuscript titled, “Hearing loss and cognitive impairment: epidemiology, common pathophysiological findings, and treatment considerations.”

We are grateful for the opportunity to submit an updated manuscript, and we have revised the paper in accordance with these valuable recommendations.  We thank the reviewers for their service and expertise, which has allowed us to materially improve the content and clarity of the submission.

We have addressed all critiques and discussed all changes in the responses below in bolded Italic. We used track change to show our in-text edits, and we have uploaded both a revised and a clean copy of the manuscript.

Reviewer 2

This is a review on the correlations between hearing loss and cognitive impairment. The manuscript is well written.

Thank you for your positive comments.

1) Better define the differences between peripheral and central age related HL.

We added a specific paragraph and relevant references to discuss the differences between central and peripheral HL. Moreover, we added details about the different origins of peripheral SNHL.

2) Better define cognitive impairment and differentiate between Dementia, mild cognitive impairment and subjective memory complain.

Thanks for this comment. We added to the revised manuscript all necessary information to understand the differences among cognitive impairment and we clarified that in this paper we generally discuss aspecific cognitive decline.

3) Adjourn the references and take in consideration for example the following papers

1: Sardone R, Castellana F, Bortone I, Lampignano L, Zupo R, Lozupone M, Griseta C, Dibello V, Seripa D, Guerra V, Donghia R, Logroscino G, Solfrizzi V, Quaranta N, Ferrucci L, Giannelli G, Panza F. Association Between Central and Peripheral Age-Related Hearing Loss and Different Frailty Phenotypes in an Older Population in Southern Italy. JAMA Otolaryngol Head Neck Surg. 2021 Jun 1;147(6):561-571. doi: 10.1001/jamaoto.2020.5334. PMID: 33570584; PMCID: PMC7879383.

2: Sardone R, Battista P, Donghia R, Lozupone M, Tortelli R, Guerra V, Grasso A, Griseta C, Castellana F, Zupo R, Lampignano L, Sborgia G, Capozzo R, Bortone I, Stallone R, Fiorella ML, Passantino A, Giannelli G, Seripa D, Panza F,Logroscino G, Quaranta N. Age-Related Central Auditory Processing Disorder, MCI, and Dementia in an Older Population of Southern Italy. Otolaryngol Head Neck Surg. 2020 Aug;163(2):348-355. doi: 10.1177/0194599820913635. Epub 2020 Apr 21. PMID: 32312167.

3: Lozupone M, Sardone R, Donghia R, D'Urso F, Piccininni C, Battista P, Di
Gioia I, Resta E, Castellana F, Lampignano L, Zupo R, Bortone I, Guerra V,
Griseta C, Seripa D, Solfrizzi V, Giannelli G, Quaranta N, Logroscino G, Bellomo A, Panza F. Late-onset depression is associated to age-related central auditory processing disorder in an older population in Southern Italy. Geroscience. 2021 Apr;43(2):1003-1014. doi: 10.1007/s11357-020-00290-1. Epub 2020 Oct 31. 

We read with interest your papers and we included the ones found appropriate to this revised version of the paper. Thank you for your suggestion.

Reviewer 3 Report

This is a comprehensive well written review describing the relation between age related hearing loss and cognitive decline.

As expected from a review, it does not add significant data to the literature, but intelligently and systematically describes what is known to date.

I believe the review warrants publication after the minor comments below are addressed.

Minor comments:

  1. While in figure 1 it is indicated that the review was based on 88 studies, in the text (page 4, line 90) it is mentioned that 95 articles were included. Could the authors explain or rectify this discrepancy.
  2. There is confusion regarding abbreviations of the imaging modalities in section 4.2 and figure 3. The full name and abbreviation should appear once before using the abbreviation only.
  3. Minor style comment: the word "nowadays" starts two sections in the text (4 and 4.2), I would use an alternative word in one of the occurrences.

Author Response

Thank you for the careful review of our manuscript titled, “Hearing loss and cognitive impairment: epidemiology, common pathophysiological findings, and treatment considerations.”

We are grateful for the opportunity to submit an updated manuscript, and we have revised the paper in accordance with these valuable recommendations.  We thank the reviewers for their service and expertise, which has allowed us to materially improve the content and clarity of the submission.

We have addressed all critiques and discussed all changes in the responses below in bolded Italic. We used track change to show our in-text edits, and we have uploaded both a revised and a clean copy of the manuscript.

Reviewer 3

This is a comprehensive well written review describing the relation between age related hearing loss and cognitive decline.

As expected from a review, it does not add significant data to the literature, but intelligently and systematically describes what is known to date.

I believe the review warrants publication after the minor comments below are addressed.

Thank you for your positive comments.

Minor comments:

1) While in figure 1 it is indicated that the review was based on 88 studies, in the text (page 4, line 90) it is mentioned that 95 articles were included. Could the authors explain or rectify this discrepancy?

Because we substantially modified the article, we also revised the number of papers included. We fixed the wrong number which was present in the previous version of the article.

2) There is confusion regarding abbreviations of the imaging modalities in section 4.2 and figure 3. The full name and abbreviation should appear once before using the abbreviation only.

We agree with your comment. The previous image was not appropriate. We now used an image that support the info we discussed in the review.

3) Minor style comment: the word "nowadays" starts two sections in the text (4 and 4.2), I would use an alternative word in one of the occurrences.

We changed the word “nowdays” with more appropriate “today” in one occurrence.

Round 2

Reviewer 2 Report

The requested changes have been done